# X-PERICL: An Explainable Method for Personality Assessment based on Hybrid Linguistic Features and In-Context Learning LLMs

## Abstract

This work proposes a novel eXplainable method for **PER**sonality assessment by linguistic features with **I**n-**C**ontext **L**earning LLMs – X-PERICL. It allows predicting the Big Five Personality traits (PTs) by text using a hybrid feature fusion combined with off-the-shelf Large Language Model (LLM). The aim is to improve both the performance and interpretability of the Personality Assessment (PA) models. Transformer-based deep features capture local contextual patterns, while hand-crafted features obtained using the Linguistic Inquiry and Word Count (LIWC) dictionary provide global and local insights into PTs reflected in text. These explainable global and local patterns are used as prompts for LLM to generate final predictions of PTs and explanations. Experiments on the ChaLearn First Impressions v2 corpus demonstrate that the integration of hand-crafted features with deep embeddings outperforms standalone representations, achieving a mean accuracy (mACC) of 0.891 and a Concordance Correlation Coefficient (CCC) of 0.333. The interpretability analysis reveals the linguistic patterns associated with each PTs, offering insights for psychological and computational linguistic research, including paralinguistics. In turn, interpretable global and local patterns based on hybrid feature fusion used as prompts for LLM enable a relative increase in Concordance Correlation Coefficient (CCC) to 9.6%. X-PERICL improves both performance and interpretability in PA, with potential applications in psychological profiling, employee selection, and personalized recommendations. The codes will be available after the paper acceptance.

## 1 Introduction

Personality traits (PTs) are stable patterns of thoughts, feelings, and behavior that distinguish individuals. The Big Five personality model, also called OCEAN, is widely used in computer science and psychology. It includes five PTs: Openness to experience (O), Conscientiousness (C), Extroversion (E), Agreeableness (A), and Neuroticism / non-Neuroticism (N) traits, which influence various aspects of behavior (McCrae & John, 1992). Personality Assessment (PA) plays a key role in understanding human behavior and finds applications in a wide range of domains, including psychology (Nikčević et al., 2021), education (Liu et al., 2024), human resourcing (Thapa et al., 2024).

Researchers use various modalities: audio, video, and text to analyze PTs, as well as bi- and multi-modal fusion strategies that combine two or more sources of information (Escalante et al., 2020). The present work focuses exclusively on the text modality, as the availability and abundance of textual data from digital platforms, such as social media, personal messages, and professional documents, make it informative for real-world applications (Panfilova & Turdakov, 2024). Importantly, text is characterized by its verbal nature as it directly reflects an individual's cognitive or linguistic style, allowing traits to be analyzed by lexical and semantic content (Bounab et al., 2024).

Traditional text-based PA methods rely on hand-crafted lexical features, such as word counts, sentiment, or word-category associations. Such features are often extracted from well-known psycholinguistic tools such as Linguistic Inquiry and Word Count (LIWC) (Pennebaker et al., 2015). Methods based on hand-crafted features can explain which words are associated with PTs, offering interpretable models to understand the relationships between linguistic patterns and PTs (Jiang et al.,

2025). However, such methods are less reliable than methods based on deep features that capture contextual and semantic nuances (Han et al., 2023). Methods based on deep features, in particular those using pre-trained language models, such as Bidirectional Encoder Representations from Transformers (BERT)-based and Word2Vec, produce rich contextual embeddings encoding syntactic, semantic, and pragmatic information. They effectively capture complex linguistic patterns in PT (Yang et al., 2021; Wang et al., 2025; Müller & Degaetano-Ortlieb, 2025).

To embrace both deep and hand-crafted features, temporal models, for example, Recurrent Neural Networks (RNNs), Long Short-Term Memory (LSTM), and Transformer, are commonly used (Bounab et al., 2024; Wang et al., 2025). These models capture sequential dependencies in textual data and analyze how linguistic patterns unfold over time. Despite the progress achieved in PA, current methods still rely on a single set of features, which limits the interpretability of the model. To overcome this limitation, such interpretation methods as SHapley Addictive exPlanations (SHAP), Local Interpretable Model-agnostic Explanations (LIME) or Large Language Model (LLM) are used (Sun et al., 2024; Wen et al., 2024; Maharjan et al., 2025; Li et al., 2025). However, SHAP and LIME do not support all model architectures, while off-the-shelf Large Language Models (LLMs) require fine-tuning. In-Context Learning (ICL) (Park et al., 2025; Wu et al., 2025) offers a flexible alternative to fine-tuning for personality inference (Sandhan et al., 2025; Handa et al., 2025), though performance remains sensitive to phrasing and prompt design.

This paper aims to address these research gaps and improve the performance of PA by developing a hybrid method based on deep feature and hand-crafted feature analysis components. To capture linguistic patterns over time, each component utilizes the Bidirectional LSTM (BiLSTM) and Mamba (Gu & Dao, 2023) models with an attention mechanism. In addition, we present an interpretation component that employs attention weights for words / tokens and word categories obtained by computing the gradients of the model output relative to the input data. The results of the last component are used as a prompt for the LLM model. The three components improve the model performance and interpretability requiring no off-the-shelf LLM training for PA.

Our main contributions are as follows:

- An explainable method X-PERICL for PA through hybrid fusion of linguistic features.
- Explainable outputs produced by X-PERICL are used as prompts to guide LLM-based predictions.
- A comparative analysis of twelve lightweight off-the-shelf LLMs across various ICL setups, including zero-, one-, few-shot, as well as explanation-based setups.

## 2 RELATED WORK

### 2.1 STATE-OF-THE-ART METHODS

State-of-the-Art (SOTA) text-based methods for PA are limited to single-set features, either hand-crafted or deep ones, and do not provide explainability without fine-tuning off-the-shelf LLM for a specific task. In 2017, a challenge was organized to develop PA methods using video interviews (Escalante et al., 2017); for this goal, a new corpus called ChaLearn First Impressions v2 (FIv2) was presented. Later, the same authors (Escalante et al., 2020) used the NLTK toolkit to extract eight readability measures and supplemented them with the total word count and the number of unique words in transcripts. Logistic Regression (LR) was applied to predict five PTs for each feature set. In contrast, Aslan et al. (2021) used pre-trained Embeddings from Language Models (ELMo) with Fully Connected Layers (FCLs) and achieved similar results. Suman et al. (2022) compared several word representation models such as Global Vectors (GloVe), BERT, Text-char-1 and 2, whose embeddings were fed into Convolutional Neural Network (CNN) and FCL to predict five PTs scores, achieving the performance comparable to the previous methods. Ouarka et al. (2024) and Revathi et al. (2025) also used GloVe with a combination of CNNs and FCLs, while Wang et al. (2025) extracted linguistic features using Contrastive Language-Image Pretraining (CLIP)-based (Radford et al., 2021) visual-text associations. Finally, Bounab et al. (2024) fine-tuned neural feature extractors (Word2Vec, Doc2Vec, and Facebook's FastText) and used five separate BiLSTM-based temporal models with self-attentions, each of which contains a single-task regressor that excludes correlations between traits. Additionally, they first trained the entire pipeline on job interview estimation

before adapting it to PA. Despite a high recognition accuracy, this method is based on multiple models for each trait independently, limited in terms of generalizability to new and non-English data, and lacks interpretability that affects practical applications.

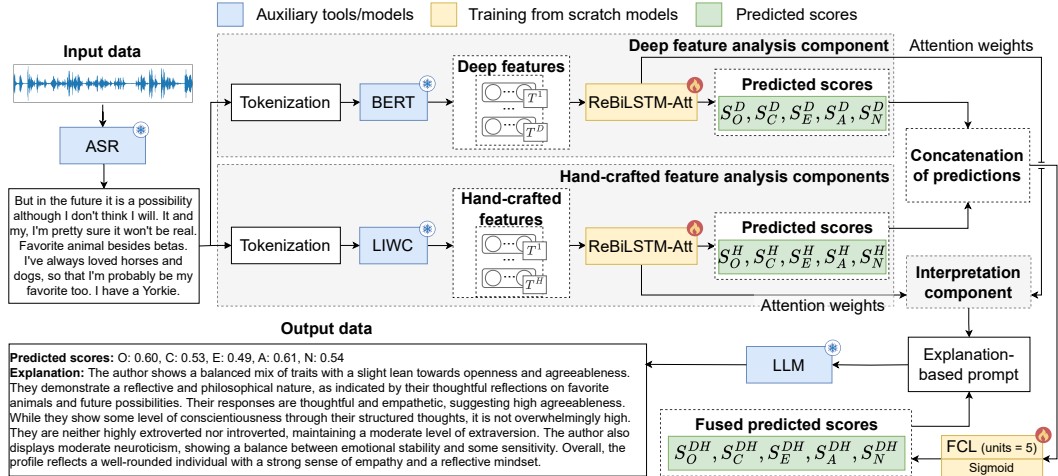

Figure 1: Pipeline of X-PERICL. $T^D$ and $T^H$ are the number of deep and hand-crafted words / tokens in an utterance, respectively.

## 2.2 INTERPRETATION METHODS

Interpretable methods are crucial for PA, given the sensitive and complex nature of PT inferences (Panfilova & Turdakov, 2024). Traditional interpretability methods, such as LIME (Ribeiro et al., 2016) and SHAP (Lundberg & Lee, 2017), are widely used in machine learning (Yang et al., 2023; Liu et al., 2025). However, these methods have notable limitations for PA. The PA models often involve high-dimensional linguistic features and multi-task learning. Zhao et al. (2022) noted that standard attention visualization tools do not provide any theoretical explanation for predictions, underscoring the need for more interpretable methods. LLMs can generate more natural and context-aware explanations for PT inferences. The idea is that LLM with its command of the language and ability to communicate in a human-like manner can articulate the reasoning behind PA model's outputs in a human-readable form. Sun et al. (2024) introduced a "chain-of-personality-evidence" framework, in which a GPT-4-based system identifies dialogue excerpts indicative of individual's traits and then composes a summary explanation connecting these clues to the Big Five prediction. Wen et al. (2024) proposed Affective Natural Language Inference (Affective-NLI), which enriches text with personality descriptions, allowing pre-trained LMs to provide explainable results. Although a gap remains to human-level explanations, integrating LLMs into PA helps narrow it.

## 3 METHODS

Figure 1 shows a pipeline of the eXplainable method for PERsonality assessment by linguistic features with In-Context Learning LLMs – X-PERICL (also refers to the famous Ancient Greek statesman Pericles). It receives a speech transcription after the Automatic Speech Recognition (ASR) by the Whisper model (Radford et al., 2023). The text is analyzed using three components: (1) deep feature analysis component; (2) hand-crafted feature analysis component; and (3) interpretation component. The first two components return five predicted scores obtained for a given input data and attention weights for them. The vectors of the predicted scores are concatenated and fed into the final FCL with five units to predict the final scores. Attention weights are used to explain: which words and word categories are more important to predict certain scores using the interpretation component. The explainability allows users to understand what patterns the model relies on, when making specific decisions on PTs. This enhances the validity of results and contributes to the confidence of professional experts. In addition, it facilitates the identification and correction of potential errors. All the components of X-PERICL are described below.

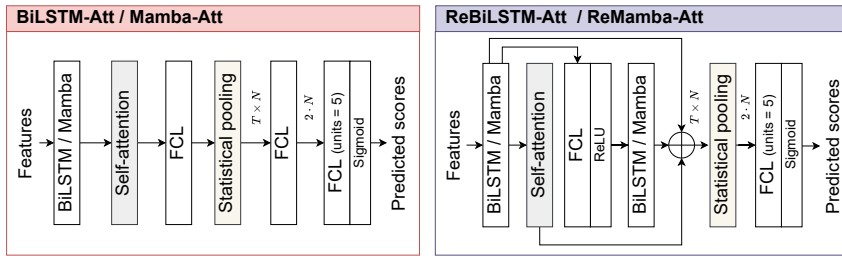

Figure 2: Proposed architectures of linguistic models: BiLSTM-Att, Mamba-Att, ReBiLSTM-Att, and ReMamba-Att. "Att" and "Re" refer to an attention mechanism and residual connections, respectively. FCL refers to Fully Connected Layer. $T$ is the number of words / (sub-)tokens in an utterance. $N$ is the number of features.

### 3.1 DEEP AND HAND-CRAFTED FEATURE ANALYSIS COMPONENTS

In the first step of both components, an input text tokenization is performed. In the deep feature analysis component, a multilingual BERT model (Devlin et al., 2019) is used to extract deep word representations (or deep features). Unknown or long words are split into sub-tokens using the WordPiece tokenization algorithm implemented in BERT. A unique identifier is assigned to each (sub-)token. The vector of unique identifiers is passed through the BERT model to extract a feature matrix with a size of $T \times 768$, where $T$ is the number of (sub-)tokens including two special tokens. The value of $T$ varies across utterances; therefore, during training, we pad sequences with zeros to the maximum length within a batch and apply a mask to ignore the padded (sub-)tokens.

In the hand-crafted feature analysis component, the Linguistic Inquiry and Word Count (LIWC) (Pennebaker et al., 2015) dictionary is used to extract the features of English words. The domain of LIWC is a psychological text analysis. In this dictionary, words are grouped into 64 categories (such as work, cognitive processes, anxiety and so on), and each word can be represented in more than one category as a binary feature vector. Each value in the feature vector is encoded with 1 if a word belongs to certain categories of LIWC, or with 0 otherwise. These feature vectors are formed into a feature matrix with a size of $T \times 64$. For this feature set, a length-based mask is also applied in the same manner as for the previous one.

We consider four models for contextual modeling of individual PTs. The proposed linguistic models are shown in Figure 2. All architectures include temporal layers, a self-attention (Vaswani et al., 2017) mechanism, and a statistical pooling layer. We use BiLSTM and Mamba (Gu & Dao, 2023) to learn context from the feature representations. The former captures sequential dependencies through gated recurrent mechanisms with bidirectional context integration. The latter, in contrast, uses a selective state-space mechanism with linear-time inference. Both models are efficient at processing long sequences. The self-attention mechanism enhances the focus of the model on informative parts of the input sequence. Statistical pooling aggregates sequential data while preserving the mean and standard deviation of feature variations. Additionally, ReBiLSTM-Att and ReMamba-Att utilizes a residual connection that improves the gradient flow during training.

The deep feature analysis component is based on pre-trained BERT models and focuses on producing rich feature representations trained on large-scale data. In contrast, the hand-crafted feature analysis component relies on predefined linguistic rules and domain-specific knowledge embedded in LIWC, which organizes text into structured categories based on linguistic and psychological attributes. Thus, deep features study non-linear relationships between words, while hand-crafted features study psycholinguistic patterns.

### 3.2 INTERPRETATION COMPONENT

We use the attention weights of proposed models for interpretation, which allows analyzing the contribution of words / tokens and word categories to the prediction of five PTs. Since LIWC groups words into 64 categories, we can evaluate the importance of a word category both locally (within an utterance) and globally (between utterances). In contrast, deep feature models assess the word importance independently. Unlike previous works (Sun et al., 2024; Bounab et al., 2024)

suggesting that word importance for each PT is proportional to frequency, our study demonstrates (see Appendix A.1) that the model attention should be prioritized over the word frequency. However, even this interpretation fails to reveal patterns for each PT due to the large lexical diversity in training data. Therefore, we use local attention weights (heatmaps) for models based on both deep and hand-crafted features and global heatmaps only for the model based on hand-crafted features.

The importance of words / tokens or word categories is calculated through the gradient of the model's output (multi-task regression) with respect to the input features, after passing through the layers of the neural network. This method for analyzing feature importance is known as Grad-CAM (Selvaraju et al., 2017). In this work, we adapt Grad-CAM to BiLSTM- and Mamba-based models to obtain heatmaps over tokens and LIWC categories.

The feature map (heatmap) construction process is almost identical for all proposed models. The models aim to solve the function $f : \mathbb{X} \to \mathbb{S}$, where $\mathbb{X}$ is the input utterance and $\mathbb{S} \in \mathbb{R}^5$ represents the predicted scores for PTs. We denote the model output by $\boldsymbol{S} = [S_O, S_C, S_E, S_A, S_N] \in \mathbb{S}$. In the initial stage, the input utterance is tokenized, and two feature matrices, $\boldsymbol{F}^D$ and $\boldsymbol{F}^H$, are extracted. $\boldsymbol{F}^D = [\boldsymbol{f}_1^D, \ldots, \boldsymbol{f}_{T^D}^D]^\top \in \mathbb{R}^{T^D \times N^D}$ refers to the deep feature matrix, where $\boldsymbol{f}_t^D \in \mathbb{R}^{N^D}$ is the deep feature vector for the $t$-th (sub-)token, $T^D$ is the number of (sub-)tokens in the input utterance, $N^D$ is the number of deep features for each (sub-)token. $\boldsymbol{F}^H = [\boldsymbol{f}_1^H, \ldots, \boldsymbol{f}_{T^H}^H]^\top \in \mathbb{R}^{T^H \times N^H}$ refers to the hand-crafted feature matrix, where $\boldsymbol{f}_t^H \in \mathbb{R}^{N^H}$ is the hand-crafted feature vector for the $t$-th token, $T^H$ is the number of tokens in an input utterance, $N^H$ is the number of hand-crafted features for each token.

The model trained on deep features receives $\boldsymbol{F}^D$ as input and produces $\boldsymbol{S}^D \in \mathbb{R}^5$. For each trait $k \in 1, \ldots, 5$, a heatmap is calculated as the partial derivative of the model output with respect to the input feature matrix:

$$\boldsymbol{FM}_k^D = \frac{\partial S_k^D}{\partial \boldsymbol{F}^D} \in \mathbb{R}^{T^D \times N^D} \tag{1}$$

where $\boldsymbol{FM}_k^D$ denotes the gradients of the $k$-th predicted trait $S_k^D$ with respect to the deep feature matrix $\boldsymbol{F}^D$. Zero-valued rows in $\boldsymbol{FM}_k^D$ are excluded from further analysis. Since deep features are not directly interpretable, we aggregate the gradients to obtain token-level importance scores. First, we average the gradient matrix $\boldsymbol{FM}_k^D$ across the token dimension, and then project the resulting vector back onto the input matrix:

$$\overline{\boldsymbol{FM}}_k^D = \frac{1}{T^D} \sum_{i=1}^{T^D} \boldsymbol{FM}_k^D[i, :] \in \mathbb{R}^{N^D}, \widetilde{\boldsymbol{FM}}_k^D = \frac{1}{N^D} \boldsymbol{F}^D (\overline{\boldsymbol{FM}}_k^D)^\top \in \mathbb{R}^{T^D}, \tag{2}$$

The resulting vector $\widetilde{\boldsymbol{FM}}_k^D$ assigns an importance value / attention weight to each (sub-)token for trait $k$. After computing $\widetilde{\boldsymbol{FM}}_k^D$, we combine sub-tokens into words and average the values for identical tokens within an utterance; the results obtained are used for visualization and analysis. The model trained on hand-crafted features uses $\boldsymbol{F}^H$ as input and produces predictions $\boldsymbol{S}^H \in \mathbb{R}^K$. For each trait $k$, we compute the gradient of the model output $S_k^H$ with respect to the input matrix $\boldsymbol{F}^H \in \mathbb{R}^{T^H \times N^H}$, resulting in a gradient matrix $\boldsymbol{FM}_k^H \in \mathbb{R}^{T^H \times N^H}$. These token-level gradients are then aggregated by averaging across the token dimension:

$$\overline{\boldsymbol{FM}}_k^H = \frac{1}{T^H} \sum_{i=1}^{T^H} \boldsymbol{FM}_k^H[i, :] \in \mathbb{R}^{N^H}. \tag{3}$$

Unlike deep features, hand-crafted features are interpretable. However, some features may not be present in a given utterance. To mask such cases, we compute a presence indicator:

$$\overline{\boldsymbol{F}}^H = \sum_{i=1}^{T^H} \boldsymbol{F}^H[i, :] \in \mathbb{R}^{N^H}, \tag{4}$$

where $\overline{\boldsymbol{F}}^H$ is the summary vector of the hand-crafted features, $\overline{\boldsymbol{F}}^H[j] = 0$ implies that the $j$-th word category does not appear in the utterance. The final masked importance vector is then defined as:

$$\widehat{\boldsymbol{FM}}_k^H[j] = \begin{cases} \overline{\boldsymbol{FM}}_k^H[j], & \text{if } \overline{\boldsymbol{F}}^H[j] \neq 0 \\ 0, & \text{otherwise} \end{cases} \quad \text{for } j = 1, \ldots, N^H. \tag{5}$$

Feature heatmaps $\widehat{\boldsymbol{FM}}_k^D$ and $\widehat{\boldsymbol{FM}}_k^H$ reflect the local feature importance of (sub-)tokens and word categories for each utterance and trait $k$, respectively. To determine the global feature importance of word categories, we average the masked hand-crafted importance vectors over the Train subset:

$$\widehat{\boldsymbol{GFM}}_k^H = \frac{1}{M} \sum_{m=1}^{M} \widehat{\boldsymbol{FM}}_k^{H(m)} \in \mathbb{R}^{N^H}, \tag{6}$$

where $M$ is the number of utterances in the Train subset, and $\widehat{\boldsymbol{FM}}_k^{H(m)}$ is the masked importance vector for trait $k$ computed on the $m$-th utterance.

This process highlights the most important (sub-)tokens and word categories that the model focuses on when predicting the five PTs. These insights provide a better understanding of the model behavior and help experts evaluate its reliability in making predictions.

### 3.3 LLM integration

The interpretation produced by our hybrid feature-based model still requires clarification and humanization of the results. To address this issue, we employ off-the-shelf LLMs. We evaluate LLMs independently of our method in three different ICL experimental setups:

1. Zero-shot, in which the model predicts scores on five PTs using only a single test utterance;
2. One-shot, in which the model receives one test and one annotated utterances;
3. Few-shot, in which the model makes predictions based on five annotated utterances.

In the final setup, called the explanation-based setup, we evaluate the most efficient LLM in combination with our hybrid feature-based method, using a generated explainable output of the method as a prompt LLM. LLM only makes predictions for the Test subset, while annotated utterances from the Train subset are used. We also use the default settings for all used off-the-shelf LLM.

## 4 Experiments

### 4.1 Research corpus

In this work, we experiment with the large-scale multimodal FIv2 corpus, which comprises 10K short high-definition video clips from YouTube with about 3K unique individuals. Each video is in English and lasts about 15 seconds. Each video is annotated with scores for each PT ranging from 0 to 1. The corpus is divided into three main subsets: Train (6K videos), Development (2K videos), and Test (2K videos) ones. The distribution of PTs scores in utterances across the subsets (see Appendix A.2) shows that the scores for four out of five PTs (O, C, A, N) are biased between 0.4 and 0.6, while the scores for the E trait are biased between 0.3 and 0.5. This bias could negatively affect the performance of PA (Escalante et al., 2020). The corpus ensures diversity by including persons of varying genders, ages, and ethnicities. This broad demographic representation, combined with a variety of monologue topics, enhances the real world applicability of models trained on FIv2.

There are some text-only corpora, including the Essays corpus (Tausczik & Pennebaker, 2010) (2500 student essays with self-reported PTs), myPersonality (Park et al., 2015) (66K Facebook users linked to 22M status updates and questionnaire scores), PANDORA (Gjurković et al., 2021) (3M Reddit comments from 1.6K users with Big-Five scores) and dialogue-based corpora, such as Story2Personality (Sang et al., 2022) (3543 characters extracted from 507 movie scripts with dialogues and scene descriptions) and PersonalityEvd (Sun et al., 2024) (about 2K dialogues of 72 users), and they remain valuable benchmarks. However, their domains are different from spoken interviews and the relying on these text-only corpora would misalign our research.

## 4.2 EXPERIMENTAL SETUP

The experimental study was conducted in several stages. Firstly, the best multilingual models for deep word representations are selected, including BERT (Devlin et al., 2019), RoBERTa (Conneau et al., 2019), and jina-embeddings-v3 (Sturua et al., 2024) (JINA). All three models are used for word representations combined with each proposed model, which are also combined with hand-crafted features obtained from LIWC. Finally, the predictions of the best model based on deep features are fused with the predictions of the best model based on hand-crafted features.

During the model training, the optimal parameters of the models and the training process are selected using the grid search method. The following parameters are considered: (1) the number of units in LSTM $\{32, 64, 128, 256\}$; (2) the number of units in FCL $\{64, 128, 256, 512\}$; (3) an optimizer $\{Adam, SGD, AdamW\}$; (4) a learning rate $\{10^{-3}, 10^{-4}, 10^{-5}\}$. For reproducibility purposes, the random seed was equal to 42. The training of the proposed models was performed using NVIDIA GeForce RTX 3090, while the testing of LLMs was performed using NVIDIA A100.

To evaluate the models, two performance measures were used: Accuracy (ACC) (Escalante et al., 2020), and Concordance Correlation Coefficient (CCC) (Lin, 1989). ACC measures an error between predicted and ground truth scores, while Concordance Correlation Coefficient (CCC) measures the correlation between them, and mean accuracy (mACC) is the average of all ACC measures for each PT (Ryumina et al., 2023).

## 4.3 EXPERIMENTAL RESULTS

Table 1 presents experimental results of proposed methods for PA based on different combinations of features (BERT, RoBERTa, JINA, and LIWC) and proposed models. It also provides results with LLMs. The mean accuracy (mACC) measure does not show significant differences in performance compared to CCC. Therefore, further performance comparisons are made by the CCC measure, while mACC is used for comparison with the SOTA methods.

In the case of deep feature representations, RoBERTa achieved the highest performance, regardless of the trained model used. The larger number of parameters in RoBERTa (278M) relative to BERT (178M) and JINA (129M) improved its ability to capture complex patterns in the feature representations, leading to superior performance. JINA (30 languages) underperformed RoBERTa (100 languages) and BERT (104 languages), probably due to more limited multilingual exposure. However, BERT's broader language coverage did not result in better performance compared to RoBERTa, suggesting that the number of parameters is more important than the diversity of languages in the feature representation for PA. The best performance was achieved using RoBERTa + ReBiLSTM-Att (mACC = 0.891; $\delta_{\text{mACC}} = 0.3\%$ relative to JINA + BiLSTM-Att); for CCC, the best result was obtained with RoBERTa + ReMamba-Att (CCC = 0.316; $\delta_{\text{CCC}} = 10.5\%$ relative to JINA + BiLSTM-Att). Models based on hand-crafted features underperformed those based on RoBERTa and BERT. LIWC + ReBiLSTM-Att achieved the highest performance with mACC = 0.889 ($\delta_{\text{mACC}} = 0.1\%$) and CCC = 0.289 ($\delta_{\text{CCC}} = 1.0\%$). Mamba-based models showed lower performance with hand-crafted features, indicating limited capabilities to model binary inputs, while when using deep features, they performed on the same level as BiLSTM.

In general, models with residual connections and two temporal layers (ReBiLSTM-Att, ReMamba-Att) outperformed their non-residual versions (BiLSTM-Att, Mamba-Att), indicating that model depth dominates over the choice of temporal layer. BiLSTM yielded higher and more stable gains than Mamba, suggesting that bidirectionality and recurrence capture long-range context more effectively than selective state-space mechanisms. Therefore, we combine the predictions of RoBERTa + ReBiLSTM-Att and LIWC + ReBiLSTM-Att in the final fusion. Combining these two methods (IDs 6 and 14) at the prediction level yielded a 7.1% relative improvement in CCC (CCC of 0.311 vs. 0.333). Thus, the fusion of deep and hand-crafted features can significantly improve the performance of the method for PA. Further analyses, including model bias and error diagnostics, are provided in Appendix A.3.

To visualize the heatmaps (importance) of the word and word categories, and to explain the model predictions (see Figure 3), we randomly selected one test utterance. The word importance for $\widetilde{\boldsymbol{FM}}_O^D$, $\widetilde{\boldsymbol{FM}}_O^H$, and $\widetilde{\boldsymbol{GFM}}_O^H$ ranges from -1 (blue, the least informative) to 1 (red, the most infor-

Table 1: Performance comparison of the proposed methods. ZS, OS, FS and EX mean zero-shot, one-shot, few-shot, and explanation-based setups. $\delta$ denotes a relative change in performance.

| ID | Method | O | C | E | A | N | mACC, % | CCC | $\delta_{\text{CCC}}$, % |
|---|---|---|---|---|---|---|---|---|---|
| 1 | JINA + BiLSTM-Att | .889 | .887 | .883 | .897 | .884 | .888 | .286 | – |
| 2 | JINA + ReBiLSTM-Att | .889 | .886 | .885 | .897 | .884 | .888 | .238 | −16.8 |
| 3 | JINA + Mamba-Att | .890 | .888 | .885 | .898 | .884 | .889 | .255 | −10.8 |
| 4 | JINA + ReMamba-Att | .891 | .887 | .885 | .897 | .884 | .889 | .276 | −3.5 |
| 5 | RoBERTa + BiLSTM-Att | .891 | .889 | .886 | .899 | .887 | .890 | .304 | 6.3 |
| 6 | RoBERTa + ReBiLSTM-Att | .892 | .889 | .887 | .900 | .888 | .891 | .311 | 8.7 |
| 7 | RoBERTa + Mamba-Att | .891 | .889 | .886 | .900 | .887 | .890 | .300 | 4.9 |
| 8 | RoBERTa + ReMamba-Att | .891 | .889 | .886 | .900 | .886 | .890 | .316 | 10.5 |
| 9 | BERT + BiLSTM-Att | .890 | .887 | .885 | .899 | .885 | .889 | .293 | 2.4 |
| 10 | BERT + ReBiLSTM-Att | .891 | .887 | .885 | .899 | .885 | .889 | .294 | 2.8 |
| 11 | BERT + Mamba-Att | .890 | .887 | .884 | .899 | .884 | .889 | .276 | −3.5 |
| 12 | BERT + ReMamba-Att | .890 | .887 | .885 | .899 | .886 | .889 | .297 | 3.8 |
| 13 | LIWC + BiLSTM-Att | .890 | .884 | .884 | .899 | .886 | .889 | .263 | −8.0 |
| 14 | LIWC + ReBiLSTM-Att | .889 | .885 | .885 | .900 | .886 | .889 | .289 | 1.0 |
| 15 | LIWC + Mamba-Att | .889 | .883 | .884 | .898 | .884 | .888 | .245 | −14.3 |
| 16 | LIWC + ReMamba-Att | .889 | .884 | .883 | .898 | .885 | .888 | .261 | −8.7 |
| 17 | IDs 6 and 14 | .892 | .889 | .887 | .901 | .888 | **.891** | .333 | 16.4 |
| 18 | Falcon-H1-7B ZS | .781 | .841 | .855 | .870 | .824 | .834 | .153 | −46.5 |
| 19 | Falcon-H1-7B OS | .853 | .838 | .857 | .866 | .826 | .848 | .172 | −39.9 |
| 20 | Falcon-H1-7B FS | .842 | .860 | .846 | .864 | .850 | .852 | .209 | −26.9 |
| 21 | ID 17 + Falcon-H1-7B EX | .887 | .885 | .882 | .899 | .886 | .888 | **.365** | 27.6 |

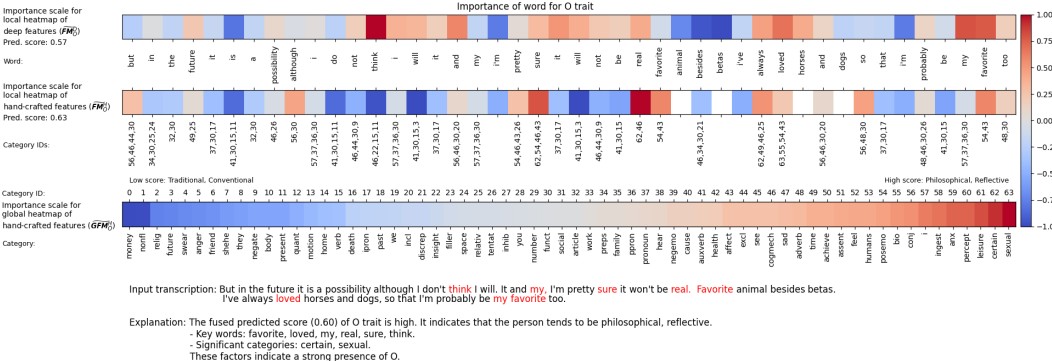

Figure 3: Visualization of importance of the word and word categories for the O trait. For $\widetilde{FM}_O^H$, each word's importance vector was computed by averaging its importance across all the categories it appeared in. Few uncolored cells indicate that a word is missing from the LIWC dictionary.

mative). For O trait, the most important words are "think", "real", "sure", "favorite", and "loved", indicating cognitive engagement, emotional expressiveness, and well-defined preferences.

These behavioral indicators are consistent with O, reflecting intellectual curiosity, self-confidence, and an appreciation for aesthetic and emotional depth. The heatmap for $\widetilde{GFM}_O^H$ illustrates word categories with values from -1 till 0 (low scores) and values from 0 to 1 (high scores). Since the text of the test utterance contains some words from highly predictive categories, such as *sexual* ("loved") and *certain* ("real" and "sure"), the predicted score for $\widetilde{FM}_O^H$ is 0.63, which is different from the predicted score for $\widetilde{FM}_O^D$ (0.57). After the fusion step, the final score is 0.6, because the model mainly relies on the $\widetilde{FM}_O^D$ prediction (see Appendix A.4). Depending on PT, the importance of words and word categories varies (see Appendix A.5). When integrating LLM into X-PERICL, the fused scores and explanations for each PT are used as prompts.

Evaluation of public lightweight off-the-shelf LLMs under different ICL experimental setups showed that not all the models can analyze prompts that adjust predictions for test utterances that

Table 2: Comparison with SOTA methods. MT refers to multi-task regressor.

| ID | Method | MT | O | C | E | A | N | mACC | CCC |
|----|--------|-----|------|------|------|------|------|------|------|
| 1 | GloVe + CNNs + FCLs (Ouarka et al., 2024) | + | .885 | .878 | .879 | .894 | .880 | .883 | – |
| 2 | NLTK + LR (Escalante et al., 2020) | + | .890 | .880 | .887 | .897 | .885 | .888 | – |
| 3 | ELMO + FCLs (Aslan et al., 2021) | + | .881 | .881 | **.901** | .893 | .885 | .888 | – |
| 4 | Text-char-2 + CNN (Suman et al., 2022) | + | – | – | – | – | – | .888 | – |
| 5 | FastText + BiLSTM-Att (Bounab et al., 2024) | – | **.904** | .886 | .886 | **.901** | .887 | **.893** | – |
| 6 | X-PERICL | + | .892 | **.889** | .887 | **.901** | **.888** | .891 | .333 |
| 7 | X-PERICL + Falcon-H1-7B | + | .887 | .885 | .882 | .899 | .886 | .888 | **.365** |

results in lower performance measures. Detailed LLMs descriptions, prompts, and full results are presented in Appendix A.6. In Table 1, we present the top three LLM-based results across zero-shot (ZS), one-shot (OS), and few-shot (FS) setups based on the highest average of the mACC and CCC measures. In the explanation-based (EX) setup, the top three positions were held by Falcon-family models with comparable performance. Therefore, we report the results of the best Falcon model. In all experimental setups, Falcon-H1-7B consistently demonstrated stable and high effectiveness. In the EX setup, this model underperformed our model on mACC by a relative value of 0.3% (0.891 vs. 0.888) and outperformed it on CCC by a relative value of 9.6% (0.333 vs. 0.365). These results show that the current LLMs cannot effectively perform PA without detailed and precise explanations that highlight important words.

Table 2 summarizes the performance of X-PERICL and the SOTA methods. In contrast to SOTA (Bounab et al., 2024), which used five single-task models, our method: (1) uses a multilingual feature extractor, which is crucial for text analysis; (2) has a single multi-task model for all PTs; (3) offers a better interpretability of results and a wider applicability.

## CONCLUSIONS

This work introduces a novel hybrid explainable linguistic method for Big Five PAs from the text data, called X-PERICL. It combines hand-crafted psycholinguistic features with deep word representations. The hand-crafted features are based on LIWC word categories, and deep features are extracted using the multilingual RoBERTa model and serve as input for ReBiLSTM-Att, which uses residual connections, two BiLSTM layers, self-attention, and statistical pooling. The intermediate predictions of both models are combined by FCL for the weighted fusion. In addition, attention weights and intermediate predictions are used to explain the model's decisions through interpretable feature heatmaps. They provide the identification of important words and word categories for each PT, producing unique explanations used with fused predicted scores as prompts for off-the-shelf LLM. Based on this, LLM predicts the final scores with a human-like explanation. Experiments on the FIv2 corpus showed that X-PERICL, independent of off-the-shelf LLM integration, outperforms the multi-task SOTA methods. Of the twelve lightweight LLMs considered, only Falcon-H1-7B demonstrated effectiveness in the four experimental setups. Using this LLM and our explanation-based prompt, X-PERICL achieved new SOTA results with CCC of 0.665. Thus, X-PERICL improves both performance and interpretability in PA, making it valuable for applications in psychological profiling, employee selection, and personalized recommendations, where understanding the reasons behind the predictions is crucial.

## LIMITATIONS

X-PERICL has a few non-critical limitations at explaining the Big Five Personality Assessment (PA). Firstly, our model is trained only on short 15-second monologues from the ChaLearn First Impressions v2 (FIv2) corpus, which may not fully represent natural conversational dynamics. Secondly, we recognize that the English-only Linguistic Inquiry and Word Count (LIWC) dictionary limits cross-lingual applications. A multilingual extension would require the use of LIWC dictionaries for other languages. Thirdly, an integration with Large Language Model (LLM) significantly increases computational costs; processing time increases from 12 to over 60 minutes for 2K utterances, making real-time applications currently impractical. Finally, the ground truth labels of FIv2 are based on observer ratings rather than standardized self-reports that can introduce a bias.

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

# A APPENDIX

## A.1 GLOBAL DEEP-BASED WORD IMPORTANCE

A visualization of the word importance for global heatmaps of deep feature representation is presented in Figure 4. The visualization results indicate that the word importance does not correlate with its frequency in training utterances. For each PT, certain words do not appear in the lists of top-40 positive/negative words across all heatmaps (only one out of five); appear in only some of the five heatmaps; appear in all the heatmaps, albeit with varying degrees of importance. For example, the word "emotional" has the highest importance for the N trait, while it has the lowest importance (among the top-40) for O. Thus, the model distinguishes patterns for each PT, which gives a general idea of how the model behaves when a particular word is used.

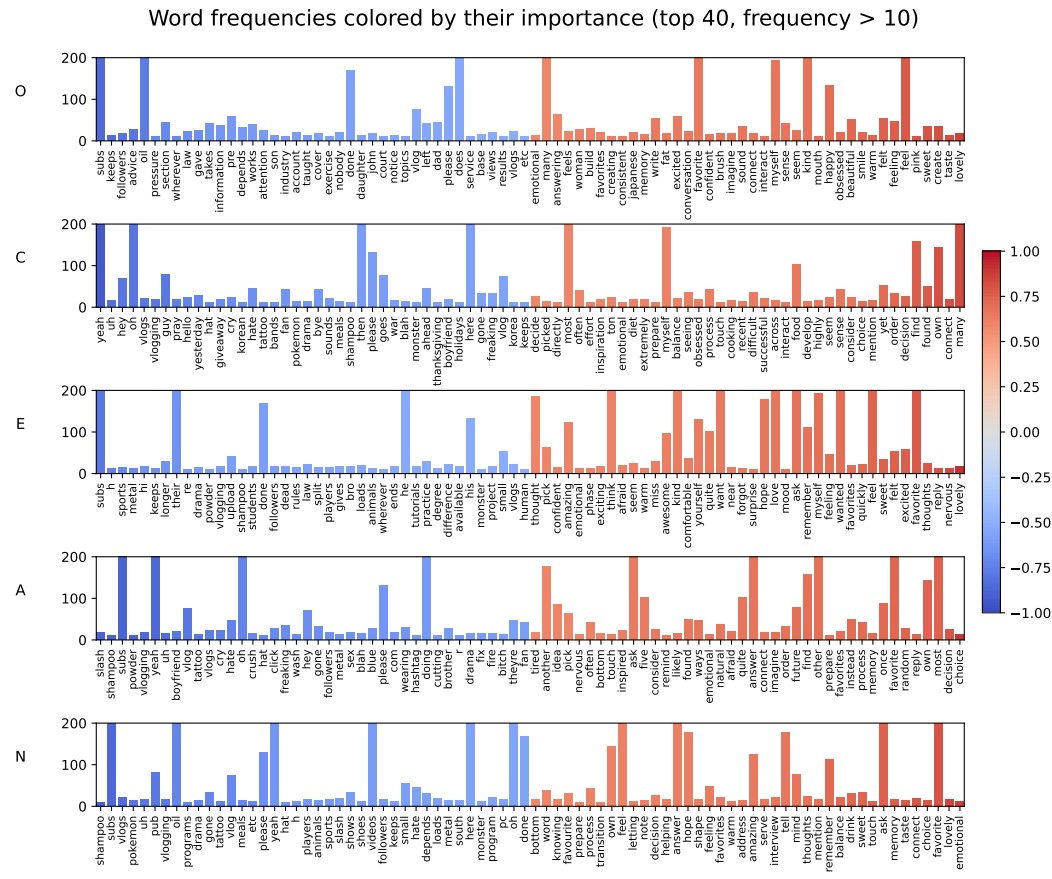

Figure 4: Visualization of word importance for global heatmaps of deep feature representation.

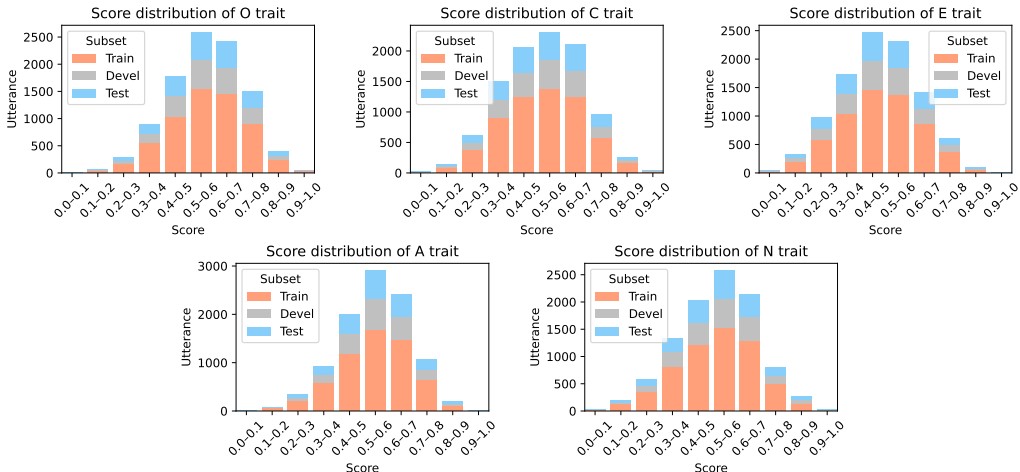

Figure 5: Distributions of PT scores in utterances across subsets.

## A.2    SCORE DISTRIBUTION IN RESEARCH CORPUS

Figure 5 shows distributions of the scores for the O, C, E, A, N traits in the Train, Development, and Test subsets. All PTs have approximately a normal distribution, with most scores in the range [0.4; 0.6]. This central tendency may cause the model to predominantly predict mean values, which may lead to reduced performance for instances with low or high PTs scores.

## A.3    STATISTICAL SIGNIFICANCE, BIAS, AND ERROR ANALYSIS

Figure 6 presents a comparison of X-PERICL with SOTA methods across the five PTs and the overall mACC. For X-PERICL, 95% confidence intervals are shown as horizontal lines with central markers indicating the estimated performance, while SOTA results are represented by single-point estimates for the same measures. X-PERICL demonstrates competitive or superior performance across all PT, with particularly strong results in C, A, and N, where its confidence interval lies above the SOTA point estimates. However, for the O and E PT, X-PERICL is inferior to SOTA (Bounab et al., 2024; Aslan et al., 2021), since the SOTA point estimates exceed the upper bounds of the corresponding confidence intervals. The mACC of X-PERICL lies within the upper range of reported SOTA results, highlighting its effectiveness in PT prediction.

The confidence intervals in terms of ACC for each PT and for the mean across PTs (mACC) were constructed using bootstrap resampling method (Tibshirani & Efron, 1993). In this procedure, sub-subsets of the same size as the original Test subset were drawn with replacement (i.e., individual test instances could appear multiple times in a single resample). Performance variability was estimated based on 1,000 bootstrap replicates, and the 2.5th and 97.5th percentiles of the resulting distribution were used as the bounds of the 95% confidence intervals.

Figure 7 shows the relationship between utterance (in tokens) and both the ground truth and predicted scores for the Big Five PTs. The ground truth scores exhibit a consistent positive trend across all PTs: as the number of tokens increases, median scores tend to rise, indicating that longer utterances are systematically assigned higher PTs values. These patterns suggest an inherent length-based annotation bias and further imply that annotation may also rely on non-linguistic data (e.g., audio or video). The model's predictions also follow this trend, but with lower variability. The predicted scores cluster more tightly around the median, than the wider range observed in human annotations, suggesting that the model may underestimate the full range of human judgments. This limitation arises because the training data are concentrated in a certain range, as shown in Figure 5. The ground truth scores fall primarily between 0.4 and 0.6, with fewer examples at the extremes. Consequently, the model lacks sufficient training data for very low or very high PT levels, so it cannot accurately predict extreme values. As a result, the model appears to learn primarily PT polarity (e.g., extraverted vs. introverted), rather than nuanced continuous PT levels. This dependence of both ground truth and predicted scores on utterance length raises concerns about fairness and gener-

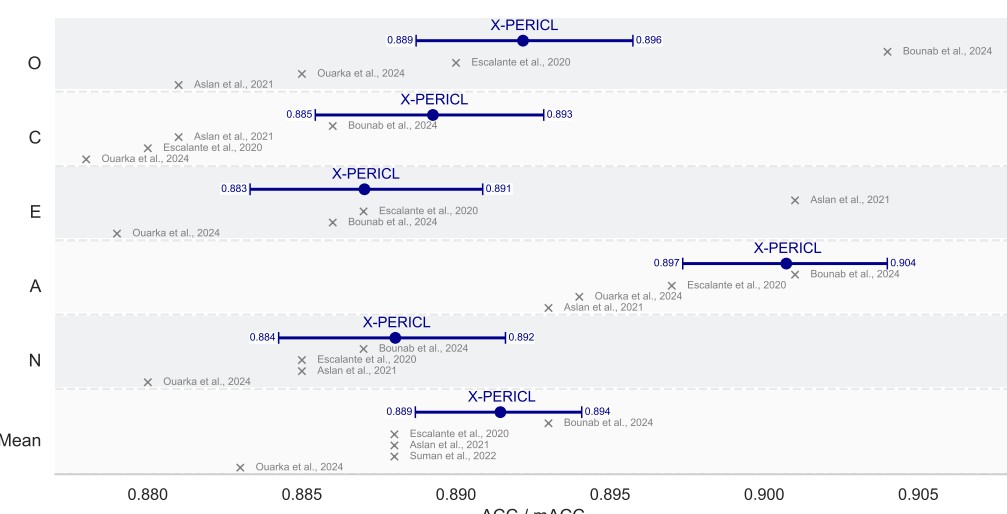

Figure 6: Comparison of the X-PERICL with SOTA. For X-PERICL, bounds with 95% confidence intervals are shown; SOTA results are represented as single-point estimates.

alization, highlighting the need to handle input length carefully in PA systems to mitigate systematic errors across different text lengths.

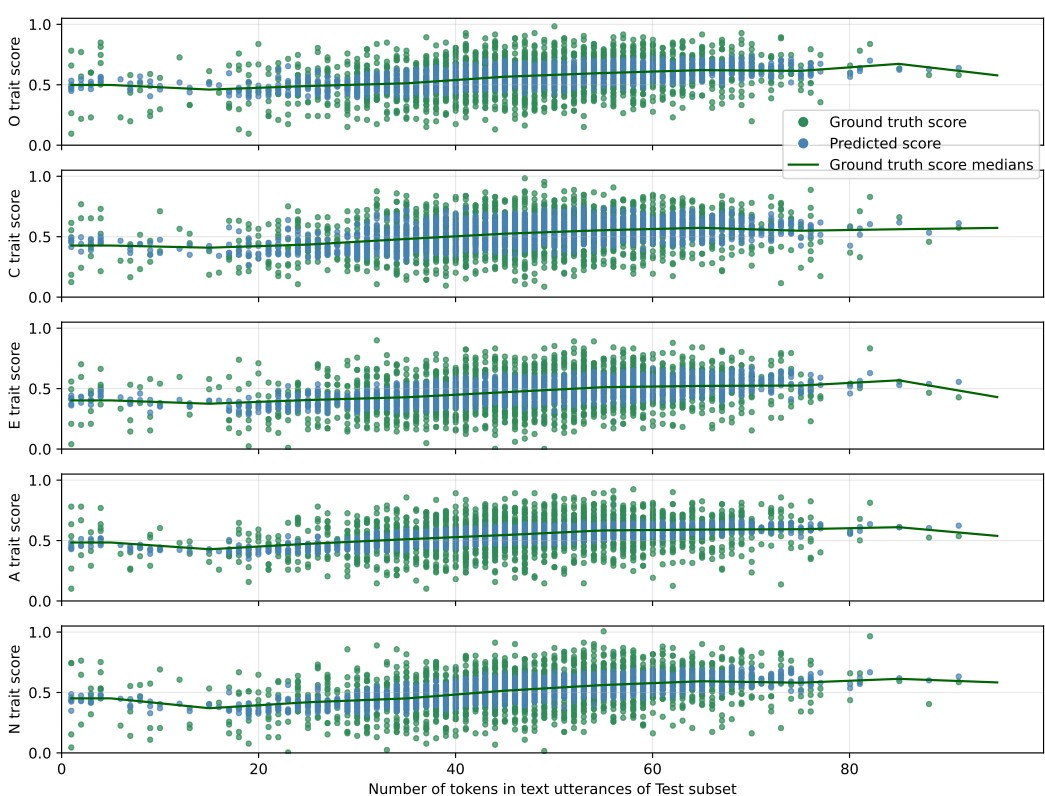

Figure 7: Comparison of predicted scores with ground truth scores across utterance lengths.

Figure 8 presents a qualitative analysis of the model's predictions using five representative examples, illustrating the interaction between ground truth scores, predicted scores, and attention patterns. The

first example demonstrates strong alignment between the ground truth and predicted scores (both low), with the model correctly identifying key linguistic cues (e.g., expressions of social withdrawal and emotional distance) that support a low E score. This suggests that the model can accurately capture relevant patterns when they are clearly expressed in the text.

The second and third examples highlight model errors arising from misaligned attention. In Example 2, the model overestimates O, assigning high attention to phrases like "so supportive", "just sharing with", and "review from other" and interpreting them as signs or indicators of intellectual curiosity. However, these phrases refer to social and material exchange, not cognitive openness. In Example 3, despite a high ground truth for A, the model underestimates the score, overlooking the polite phrase "so please do not spark me up". Instead, it focuses on neutral proper names (e.g., Curry, Westbrook) and repetitive phrasing (e.g., "we are gonna go"), which together lack trait-relevant meaning. This reflects a tendency to prioritize surface-level tokens over contextually grounded linguistic signals.

Examples 4 and 5 reveal potential issues in the annotation process. In Example 4, the model predicts a high score based on clear indicators of conscientiousness (e.g., "focus on my studying", "couple months"), yet the ground truth score is low. Similarly, in Example 5, the model predicts a low N score, driven by attention to hesitation markers such as "I mean" and fragmented phrasing, which typically signal uncertainty and thus support a low N prediction; however, the ground truth score is high. These discrepancies suggest that the annotations may not fully reflect the textual content, and may instead rely on non-linguistic cues (e.g., audio or video). Thus, these cases point to annotation errors, where human estimates deviate from the evidence present in the text.

Overall, the analysis shows that the model's predictions fall into either correct or incorrect outcomes. For the latter, errors arise either from model limitations, such as misprioritizing linguistic cues or from annotation inconsistencies, wherein ground truth scores do not align with textual evidence. This distinction underscores the importance of evaluating both model performance and data quality in PA, particularly when interpreting results in real-world applications.

## A.4 SCORE CONTRIBUTION TO FINAL PREDICTIONS

Figure 9 illustrates how intermediate predictions contribute to the final PT scores. In most cases, the fusion model relies primarily on predictions based on deep features. The diagonal entries indicate that each trait depends mainly on its own intermediate prediction, with the highest contributions for O ($S_O^{DH} = 0.60$) and C ($S_C^{DH} = 0.70$). For the remaining traits, the off-diagonal values reveal cross-trait influences. E receives notable input from O ($S_O^D = 0.08$) and C ($S_C^H = 0.07$). A distributes weights almost equally between all other PT. N is moderately influenced by E ($S_E^D = 0.09$ and $S_E^H = 0.10$) and C ($S_C^H = 0.09$). The relatively low off-diagonal values suggest that the final scores are largely trait-specific, with limited interdependence across the OCEAN dimensions.

## A.5 VISUALIZATION OF HEATMAPS

The visualization of feature heatmaps (importance) of the word and word categories for traits C, E, A, N are shown in Figures 10, 11, 12, and 13. It can be seen that different words are activated in local heatmaps depending on the target PT. Moreover, models based on deep ($\widetilde{\boldsymbol{FM}}_k^D$) and hand-crafted ($\widetilde{\boldsymbol{FM}}_k^H$) feature heatmaps, where $k \in \{O, C, E, A, N\}$, differ in the nature of activation. For example, the word "think" contributes positively to $\widetilde{\boldsymbol{FM}}_k^D$ only for the E, A, and N traits, suggesting that it reflects social, empathetic, or overthinking tendencies. The word "horses" is activated exclusively for the A trait in $\widetilde{\boldsymbol{FM}}_A^D$, potentially indicating an association with empathy and affection. Notably, "horses" is absent from the LIWC dictionary, making it impossible to estimate its contribution to $\widetilde{\boldsymbol{FM}}_A^H$. This discrepancy indicates that models based on different features are complementary ones. Meanwhile, the word "favorite" appears in both $\widetilde{\boldsymbol{FM}}_k^D$ (in only one position out of two) and $\widetilde{\boldsymbol{FM}}_k^H$ (in both positions) for the E and N traits, which means that it reliably labels strong preferences for sociable or emotional people.

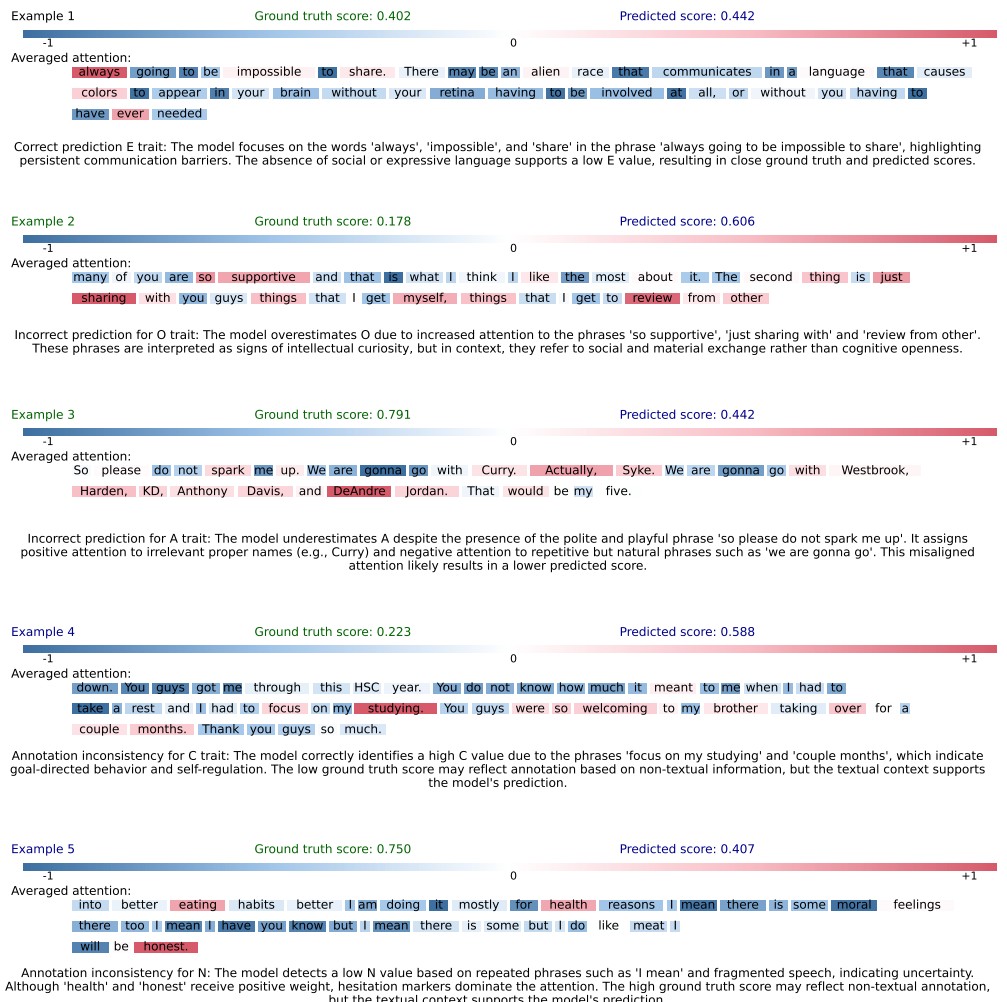

Figure 8: Qualitative analysis of five examples illustrating (1) a correct prediction, (2–3) model errors, and (4–5) annotation inconsistencies.

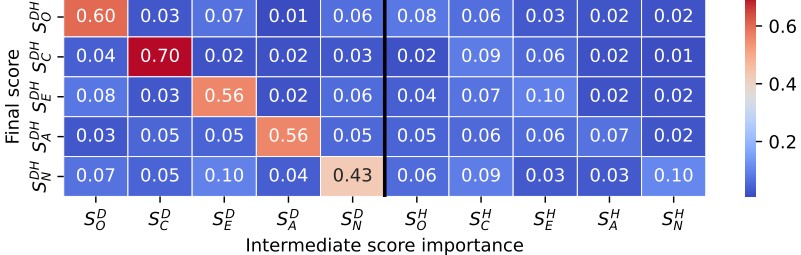

Figure 9: Contribution of intermediate predicted scores to final scores.

The importance of word categories is different for each PA in the global feature heatmaps ($\widetilde{GFM}_k^H$). Each importance value varies from -1 to 1 and defines the opposite trait (from low to high). For example, when speaking words from the categories *nonfl* (words such as "er", "hm"), *money* ("audit", "cash"), etc., the model predicts low scores for the O trait; when speaking words from the categories *sexual* ("horny", "love"), *certain* ("always", "never"), etc., the model predicts high scores for the same trait. Similar importance of word categories is presented for the E trait.

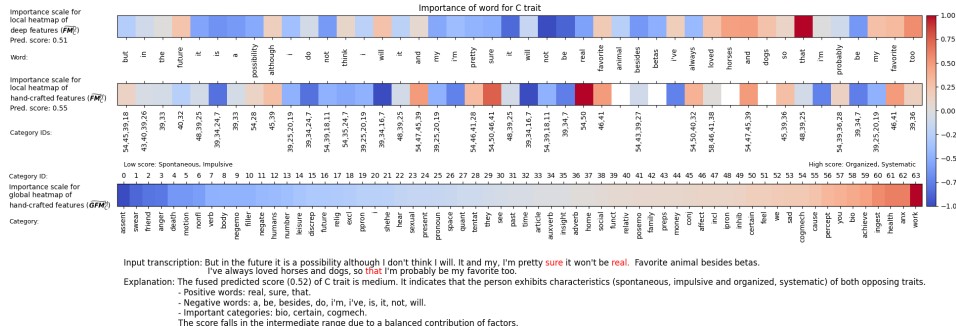

Figure 10: Visualization of importance of the word and word categories for C trait. For $\widetilde{FM}_C^H$, each word's importance vector was computed by averaging its importance across all the categories it appeared in. Uncolored cells indicate that a word is missing from the LIWC dictionary.

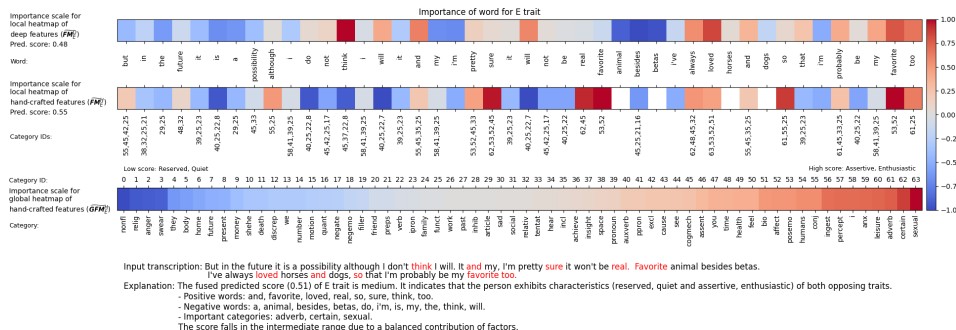

Figure 11: Visualization of importance of the word and word categories for E trait.

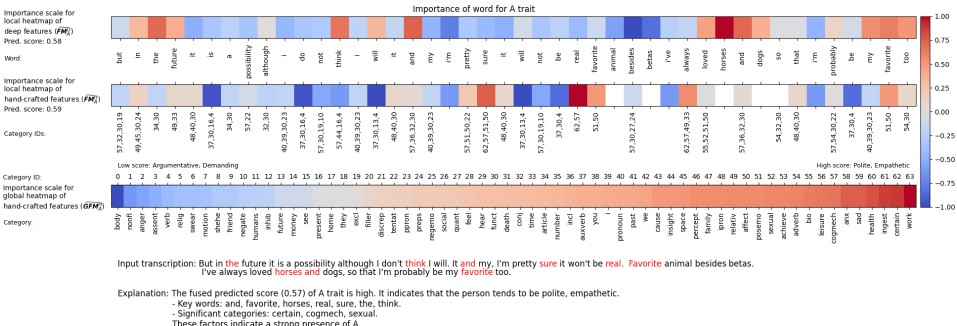

Figure 12: Visualization of importance of the word and word categories for A trait.

The use of words from such categories as *assent* ("agree", "OK", "yes"), *swear* ("damn", "sob"), etc. indicates a person's tendency to be impulsive (low scores for C); while words from such categories as *work* ("job", "majors"), *anx* ("worried", "fearful"), etc. indicate a person's tendency to be systematic (high scores for C). The similar importance of these word categories is also typical for the A trait. Finally, the N trait has a lower correlation with the other traits. For this trait, such word categories as *friend*, *nonfl*, etc. indicate a person's low emotional stability; while categories such as *percept* ("observation", "hearing"), *anx*, etc. indicate a high person's emotional stability.

## A.6 LARGE LANGUAGE MODEL RESULTS

Table 3 shows a comparison of the LLMs characteristics. We used only models with parameters less than 10B (Qwen, Falcon, Gemma, Phi, Yandex, Hunyuan) because: (1) they enable faster experiments and less resource consumption, which is important for scalable evaluations; (2) their

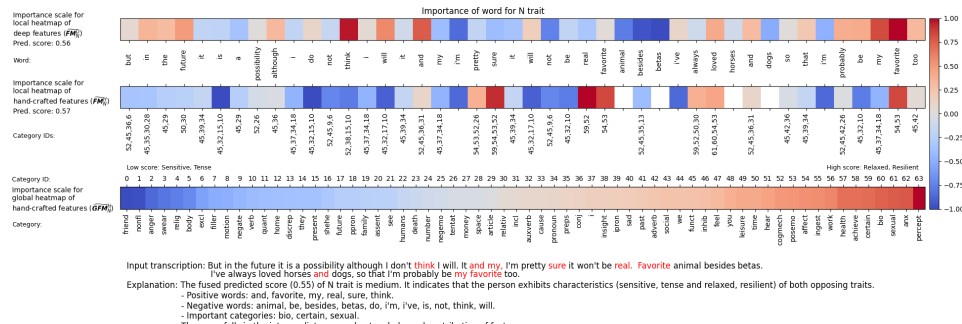

Figure 13: Visualization of importance of the word and word categories for N trait.

Table 3: Comparison of LLMs characteristics. Inference time is given for 2K test requests.

| Model | Short name | Link | Parameters | Layers | Architecture | Training Data | Inference time, min. |
|---|---|---|---|---|---|---|---|
| Qwen2.5-7B-Instruct | Qwen2.5-7B | HF | 7.61B | 28 | RoPE, SwiGLU, RMSNorm | 18T tokens across over 29 languages; includes code-related data | ≈ 50 |
| Qwen3-1.7B | Qwen3-1.7B | HF | 1.7B | 28 | LLaMA-based | 36T tokens across 119 languages; includes coding, STEM, reasoning data | ≈ 40 |
| Qwen3-4B | Qwen3-4B | HF | 4B | 36 | LLaMA-based | 36T tokens across 119 languages; includes coding, STEM, reasoning data | ≈ 60 |
| Qwen3-8B | Qwen3-8B | HF | 8B | 40 | LLaMA-based | 36T tokens across 119 languages; includes coding, STEM, reasoning data | ≈ 80 |
| Falcon3-7B-Instruct | Falcon3-7B | HF | 7B | 32 | Decoder-only | 14T tokens; supports 4 languages | ≈ 60 |
| Falcon3-10B-Instruct | Falcon3-10B | HF | 10B | 40 | Decoder-only | 14T tokens; supports 4 languages | ≈ 80 |
| Falcon-H1-7B-Instruct | Falcon-H1-7B | HF | 7B | 44 | Decoder-only, Transformers + Mamba | Supports English, Multilingual | ≈ 60 |
| YandexGPT-5-Lite-8B-instruct | Yandex-5-8B | HF | 8B | 32 | LLaMA-based | 15T tokens (web, code, math), then 320B high-quality tokens | ≈ 60 |
| Gemma-3-1B-it | Gemma-3-1B | HF | 1B | 24 | Decoder-only | 2T tokens across 140 languages from diverse sources | ≈ 90 |
| Gemma-3-4B-it | Gemma-3-4B | HF | 4B | 32 | Decoder-only | 4T tokens across 140 languages from diverse sources | ≈ 180 |
| Phi-4-mini-Instruct | Phi-4-mini | HF | 3.8B | 36 | Decoder-only | 5T tokens across 24 languages from diverse sources | ≈ 30 |
| Hunyuan-7B-Instruct | Hunyuan-7B | HF | 7B | 32 | Decoder-only | Not available | ≈ 50 |

open-access weights ensure reproducibility of inference (unlike closed APIs); and (3) they better match our benchmarking goal (non-peak performance) compared to such LLMs as ChatGPT, DeepSeek, or other proprietary LLMs. We also evaluated others LLMs not presented in Table 3, such as SmolLM3-3B and MiMo-7B-RL, but since they produced gaps in their predictions, we excluded them from the comparison.

The selection of prompts is an important step in testing LLMs. Figure 14 presents the prompts that show the highest average performance among all the models. However, it should be noted that some models performed better on certain prompts, while others excelled on different ones. Since the goal of this work is not to optimize the performance of LLMs, but to demonstrate their ability in PA without fine-tuning for a specific task, we limit our analysis to the most consistently effective prompts.

Experimental results (see Figure 15) show that including prompts in the one-shot and few-shot setups generally improves the performance of off-the-shelf LLMs. In particular, heavier models respond worse to prompting compared to lighter architectures, suggesting that lighter models may benefit more from structured guidance and task-specific prompts. Our method for generating explanation-based prompts for LLMs significantly improves performance: mACC increases by 0.360 (0.891 vs. 0.531), while CCC increases by 0.360 (0.365 vs. 0.005).

Therefore, lightweight models benefit disproportionately from our explanatory prompts, compensating for limited pre-training. This inverse size-utility relationship enables a performant and resource-efficient deployment through prompt engineering alone.

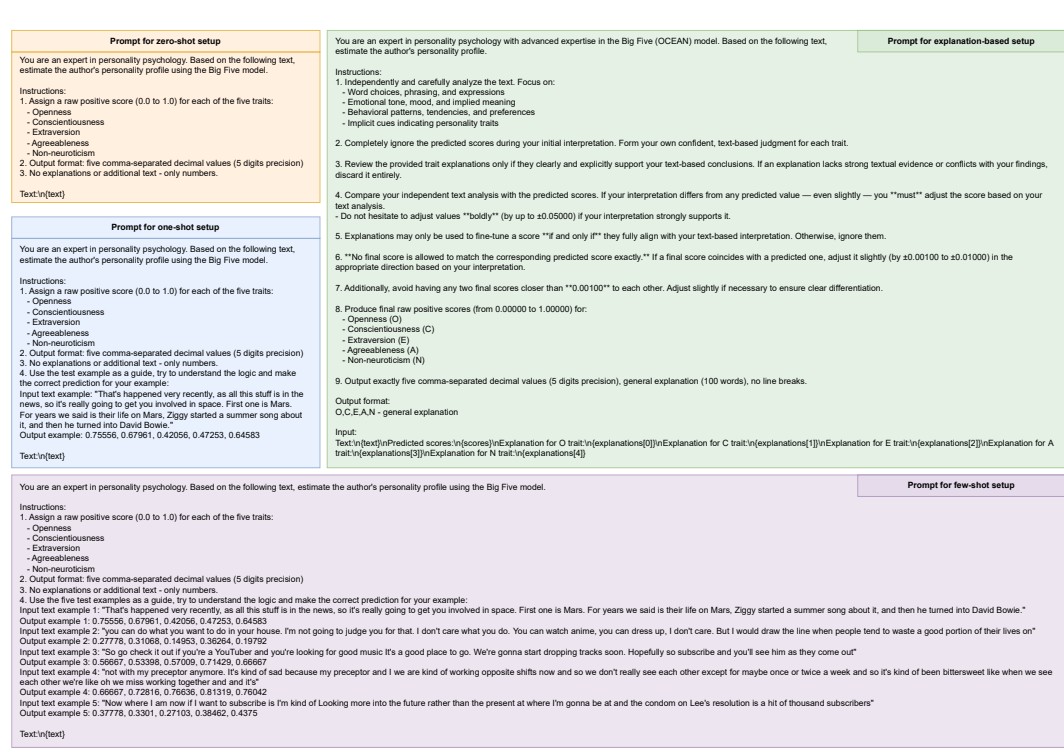

Figure 14: Prompts for testing of LLMs.

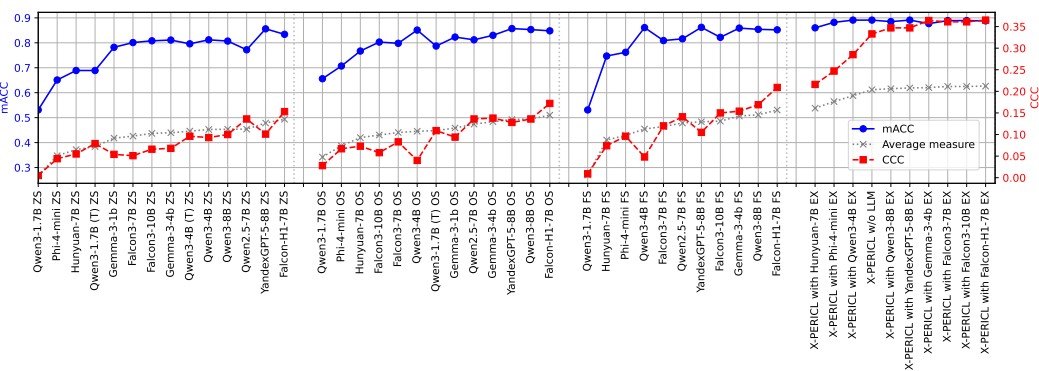

Figure 15: Performance measures of LLMs. ZS, OS, FS and EX refer to zero-, one-, few-shot and explanation-based setups. T means a thinking mode.

