# OpenReview forum: "X-PERICL: An Explainable Method for Personality Assessment based on Hybrid Linguistic Features and In-Context Learning LLMs"
_ICLR.cc/2026/Conference — ICLR 2026 Conference Withdrawn Submission_

### Official Review · Reviewer_3vGq · 2025-10-23

**Soundness:** 3
**Presentation:** 2
**Contribution:** 2
**Rating:** 4
**Confidence:** 3

**Summary:**

This paper introduces X-PERICL, a novel explainable method for personality assessment that combines hybrid linguistic features with in-context learning Large Language Models (LLMs). The approach integrates transformer-based deep features with hand-crafted LIWC (Linguistic Inquiry and Word Count) dictionary features to predict the Big Five personality traits from text. X-PERICL consists of three main components: (1) a deep feature analysis component using BERT embeddings, (2) a hand-crafted feature analysis component using LIWC categories, and (3) an interpretation component that generates explanations for predictions. The method uses BiLSTM and Mamba models with attention mechanisms to process these features, and the resulting explanations serve as prompts for off-the-shelf LLMs to generate final predictions. Experiments on the ChaLearn First Impressions v2 corpus demonstrate that the hybrid feature fusion outperforms standalone representations, achieving a mean accuracy of 0.891 and a Concordance Correlation Coefficient of 0.333, with a relative CCC increase of 9.6% when combined with LLMs.

**Strengths:**

The hybrid approach effectively combines deep features (capturing contextual patterns) with hand-crafted features (providing interpretable insights), outperforming methods that rely on a single feature type.

The method provides transparent explanations for personality predictions through attention weights and gradient-based feature importance visualization, making it more trustworthy for applications in psychology and HR.

The framework is flexible, allowing integration with various off-the-shelf LLMs without requiring fine-tuning, which enhances practical applicability.

The authors conduct extensive experiments comparing different model architectures (BiLSTM vs. Mamba) and feature extractors (BERT, RoBERTa, JINA), providing valuable insights into their relative strengths.

The approach achieves state-of-the-art performance on the ChaLearn First Impressions v2 corpus while maintaining interpretability.

**Weaknesses:**

The method relies on the English-only LIWC dictionary, limiting cross-lingual applications and requiring separate dictionaries for other languages.

The model is trained only on short 15-second monologues from a single corpus, which may not fully represent natural conversational dynamics or generalize to other text types.

Integration with LLMs significantly increases computational costs, with processing time increasing from 12 to over 60 minutes for 2,000 utterances, making real-time applications currently impractical.

The ground truth labels in the FIv2 corpus are based on observer ratings rather than standardized self-reports, which could introduce bias in the training data.

The error analysis reveals that the model struggles with extreme personality trait values, tending to predict scores in the middle range due to the distribution bias in the training data.

**Questions:**

How might the model's performance be affected when applied to longer texts or different discourse types beyond short monologues?

Could the approach be extended to capture multimodal personality cues (e.g., combining text with audio or video features) to address the limitations of text-only analysis?

How could the computational efficiency of the LLM integration be improved to make the method more suitable for real-time applications?

How would the model perform on datasets with more balanced distributions of personality trait scores, particularly for extreme values?

Could the approach be adapted to predict other psychological constructs beyond the Big Five personality traits, such as values, attitudes, or emotional states?

---

### Official Review · Reviewer_vQpV · 2025-10-27

**Soundness:** 1
**Presentation:** 2
**Contribution:** 2
**Rating:** 2
**Confidence:** 4

**Summary:**

The paper proposes X‑PERICL, a text-based method for Big‑Five personality score prediction and attribution. The method works on multi-stage pipeline:

(1) extracts language model representations and hand‑crafted features (LIWC) from speech transcript

(2) runs each sequence of features through BiLSTM/Mamba WITH attention

(3) fuses the two predicted score vectors with an fully connected layer + sigmoid

(4) uses gradients to produce token / category heatmaps that are then turned into an “explanation‑based prompt” for an LLM to rescore the traits and generate explanation

All experiments are on First Impressions v2 dataset speech transcripts. Authors claim their proposed method to outperform SOTA methods based on accuracy and concordance correlation coefficients.

**Strengths:**

- The paper proposes a modular pipeline that combines language representations (what authors call ‘deep’) and LIWC features, with visual interpretability (local token + global LIWC category heatmaps). The overall pipeline is clearly shown in Figure 1.

- Attempting to provide a natural language description of the prediction is important as it has potential to facilitate the psychology research supported with generative AI. Previous works often focus on prediction rather than attribution, while this paper attempts to tackle attribution with attention weights.

**Weaknesses:**

- The SOTA claim is not supported by the experiment data, in two ways.

1. In Table 2, a prior method (FastText+BiLSTM‑Att, row ID 5) has higher mACC and several individual personality traits’ ACCs than X‑PERICL. However, the paper claims X-PERICL prediction outperforms others. It is not just FastText+BiLSTM‑Att, but also NLTK+LR (Escalante et al.) and ELMO+FCLs (Aslan et al.) that seem to show comparable performances; to check the statistical significance I looked at Figure 6 (line 756) but statistical testing is asymmetrical (i.e. only done for X-PERICL, not for baselines) and strongly implies that nearly all baseline methods will be comparable if considering their confidence intervals.

There is also an ambiguity in the definition of evaluation metrics: please refer to the ‘Questions’ section.

2. Only a single English dataset, FI v2, is used while there are multiple spoken datasets used for personality research; for example, please refer to the Table 1 of the paper “Multimodal Personality Traits Assessment (MuPTA) Corpus” (Ryumina et al.) for the list of datasets. I acknowledge that some text corpora for personality research might be incompatible with authors’ focus on spoken language -- as mentioned in Lines 317-324 -- still, the paper's claim should be supported by multiple datasets that has audio modality data paired with personality trait.

There is also an issue regarding the language domain. In the related work (Line 109), the authors note that prior studies are “limited in terms of generalizability to new and non-English data.” However, the experiments are conducted on the FIv2, which is an English dataset. Since authors identify this as a key limitation of previous work, it would be important for the authors to demonstrate the cross-lingual generalizability of X-PERICL.

- Methodological clarity.

Under “LLM integration” (Section 3.3), it says “we also use the default settings” for all LLMs and that LLMs only predict on Test while in-context examples are drawn from Train, but how in-context examples are selected and whether they include explanations/scores is not clearly explained.

I also noticed that the actual prompts (Figure 14) include highly specialized constraints like: “Do not hesitate to adjust values **boldly** (by up to ±0.05000)” and “avoid having any two final scores closer than **0.00100** to each other” but could not find a justification for designing these instructions and specific numbers like 0.00100, nor a prompt engineering process. These instructions indicate a significant shift from using LLMs for simply refining predictions based on explanations and complicates fair comparison to previous methods.

Missing training details: the paper lists LSTM units {32–256} and FCL sizes {64-512} in Line 334, but omits loss function (MSE / MAE / CCC?), batch size, training steps, regularization, etc. Also, one ambiguous part I noticed is the mismatch in hidden dimensions – BERT features are 768-dimensions, but authors tested LSTM units 32-256, and in Figure 2 BERT features are directly fed to LSTM which causes mismatch in dimensions. The absence of these explicit details makes exact reproduction unlikely.

Finally, code is not provided. Considering the paper’s novel ReBiLSTM‑Att architecture and the Grad‑CAM adaptation, along with the methodological ambiguities mentioned above (selection of in-context examples, prompt engineering, details like hidden units and loss function for training) I think authors should release the codebase in advance.

- While the method generates natural language descriptions of prediction, its usage and performance is not clearly evaluated. All metrics (Table 1,2) are based on quantitative metrics like ACC, mACC. Without careful evaluation of natural language descriptions, the overall framework of X-PERICL is unreliable such as a risk of bias and stereotypes contained in the description, and undermines the main contributions (Line 84) of “Explainable outputs produced”.

**Questions:**

Inconsistent experiment result. The best CCC (the metric authors use) is 0.365 (Line 449, Table 1 row 21) but the conclusions mention new SOTA results with CCC of 0.665 (Line 471). I would like to ask authors to provide a consistent result and make revisions if this is just a simple typo.

Unclear description of the metric ACC. The paper says “ACC measures an error between predicted and ground truth scores” (line 340) which does not seem to match the problem setup where the predicted scores are real numbers in range (0,1). The formula is omitted, making readers hard to understand the metric. I would like to ask authors for detailed descriptions of experiment metrics.

Line 66: SHapley Addictive exPlanations -> should be ‘Additive’, not ‘Addictive’

Line 327: cited Conneau et al. for RoBERTa; probably authors intended to cite Liu et al. (2019) ? Conneau et al. paper is proposing XLM-R (XLM-RoBERTa).

---

### Official Review · Reviewer_3AWn · 2025-10-30

**Soundness:** 2
**Presentation:** 3
**Contribution:** 2
**Rating:** 4
**Confidence:** 4

**Summary:**

This paper introduces X-PERICIL, a method for personality assessment (along the big 5 axes) from spoken language data. Here, a short (15 second) utterance is converted to text (ASR, whisper), then passed through two branches: a deep feature analysis component, and a hand-crafted feature set. The deep-features are extracted using BERT, while the hand-crafted features are extracted using an LWIC dictionary. Both are then passed through a ReBiLSTM,  and fused to predict the personality scores. These personality scores/some of the internal state is then passed to an LLM (Falcon-H1-7B), which generates an "explainable" version of the scores (based on a reasoning prompt). The method is both trained and evaluated on FIv2 corpus (10L video clips, 3K individuals). The final method shows improvements in some category predictors.

**Strengths:**

This paper is well written, with a very strong narrative. The goal of predicting personality is important, and the authors provide a clear motivation for why existing approaches fall short. The paper is the first to take this approach, and performs on-par with existing methods, while being explainable (and maintaining a multi-task regressor).  Something that stands out, the experimental ablations are also quite comprehensive, spanning multiple feature extractors, architectural variants, and prompting setups.

**Weaknesses:**

The paper also has some weaknesses:
- There's no statistical analysis of the results, particularly in Table 1, so it's quite challenging to understand which of these numbers are significantly different than the others. It's also not clear if the small changes in mACC, for example, actually correlate to tangible differences in the predictions. The CCC metric appears to have higher variance, but this is also lacking statistical analysis (including an analysis of variance), which means that it's hard to tell which methods are truly different.
- It's not particularly clear that the model achieves strong results compared to baselines. In Table 2, mACC is on-par or equal to all prior methods, and CCC is only computed for the X-PERICL method. Interestingly, adding Falcon-H1-7B as an LM seems to make the mACC worse.
- The overall architecture isn't particularly well motivated: while the ablations are comprehensive, the full underlying architecture isn't very well motivated: why do we need a sequence model? What inspires the choice of layers in Figure 2?
- There's no evidence that this will generalize to other texts. The model could have been evaluated on some of the mentioned text datasets (For example, Essays, myPersonality, PANDORA, etc.) to give some notion of how the model generalizes. I acknowledge that there is a domain gap, but these cross-dataset performance elements would be helpful to give a picture of how well the model is able to perform in general compared to existing non-spoken SOTA. Similarly, this model could be trained on PANDORA, and evaluated on other standard text, to validate the architecture design.
- There's no quantitative evaluation of the explainable component of the model, which is probably the main contribution of the method. It would be nice to see some evaluations (either human evals, or some quantitative measurement).

**Questions:**

- How is the data split between training/testing? Do the same personalities/people appear in both train and test splits? (Given that there's 10K samples, but only 3K individuals). This could cause some generalization error, if the model has been trained on the same personalities it is evaluated on.
- How well do LLMs (or models such as Falcon-H1) perform on this task zero-shot? It feels like GPT-4 would be a strong baseline for predictive performance given the results in [1,2].

[1] Tak, Ala N., et al. "Mechanistic Interpretability of Emotion Inference in Large Language Models." arXiv preprint arXiv:2502.05489 (2025).
[2] Suh, Joseph, et al. "Rediscovering the Latent Dimensions of Personality with Large Language Models as Trait Descriptors." NeurIPS 2024 Workshop on Behavioral Machine Learning.

---

### Note · Authors · 2025-11-24

**Comment:**

We would like to thank the reviewers for their valuable feedback. While some of the suggestions, such as clarifying metrics and correcting minor errors, are relatively easy to address, others, such as evaluating on additional corpora and conducting cross-lingual experiments, require more time and resources. Due to the scope of the recommendations, we have decided to withdraw the paper and revise it.

**Withdrawal Confirmation:**

I have read and agree with the venue's withdrawal policy on behalf of myself and my co-authors.